# In Vitro Calli Production Resulted in Different Profiles of Plant-Derived Medicinal Compounds in *Phyllanthus amarus*

**DOI:** 10.3390/molecules25245895

**Published:** 2020-12-12

**Authors:** Maria Eduarda B. S. de Oliveira, Adilson Sartoratto, Jean Carlos Cardoso

**Affiliations:** 1Biotechnology Undergraduate Course of Centro de Ciências Agrárias, Universidade Federal de São Carlos (CCA/UFSCar), Rodovia Anhanguera, km 174, Araras 13600-970, Brazil; dudabarboza.oliveira@gmail.com; 2Centro Pluridisciplinar de Pesquisas Químicas, Biológicas e Agrícolas (CPQBA), Universidade Estadual de Campinas, Campinas 13083-970, Brazil; adilson@cpqba.unicamp.br; 3Department of Biotechnology, Plant and Animal Production, CCA/UFSCar, Rodovia Anhanguera, km 174, Araras 13600-970, Brazil

**Keywords:** *Phyllanthus amarus*, phyllanthin, hypophyllanthin, callus, micropropagation, secondary metabolites

## Abstract

The efficient production of plant-derived medicinal compounds (PDMCs) from in vitro plants requires improvements in knowledge about control of plant or organ development and factors affecting the biosynthesis pathway of specific PDMCs under in vitro conditions, leading to a realistic large-scale tool for in vitro secondary metabolite production. Thus, this study aimed to develop an in vitro technique, through the induction and proliferation of calli, for production of plant fresh weight, and to compare the PDMC profile obtained from the plants versus in vitro calli of *Phyllanthus amarus*. It was successfully possible to obtain and proliferate two types of calli, one with a beige color and a friable appearance, obtained in the dark using Murashige and Skoog (MS) medium plus 2,4-dichlorophenoxyacetic acid (2,4-D), and a second type with a green color, rigid consistency, and nonfriable appearance obtained under light conditions and MS medium plus 6-benzyladenine (6-BA). In vitro micropropagated plants that gave rise to calli were also acclimatized in a greenhouse and cultivated until obtaining the mass for PDMC analysis and used as a control. While the micropropagated-derived plants concentrated the lignans niranthin, nirtetralin, and phyllanthin, the *Phyllanthus amarus* calli proliferated in vitro concentrated a completely different biochemical profile and synthesis of compounds, such as betulone, squalene, stigmasterol, and β-sitosterol, in addition to others not identified by GC-MS database. These results demonstrate the possibility of applying the calli in vitro from *Phyllanthus amarus* for production of important PDMCs unlike those obtained in cultures of differentiated tissues from field plants.

## 1. Introduction

The species *Phyllanthus amarus* belonging to the family Phyllanthaceae is an annual plant with a variation between 10 and 60 cm in height with a simple main stem and which may have branches [1]. It is a cosmopolitan plant with a wide distribution in the tropics and subtropics, as well as in the American continent, occurring mainly in Brazil and the Caribbean [2]. It is a medicinal plant popularly known as stonebreaker, as it has historically been used in the elimination of kidney stones, commonly used as an herbal medicine [3]. Its clinical effects include antihepatotoxic, antilytic, antihypertensive, anti-inflamatory, anti-human immunodeficiency virus (HIV), and anti-hepatitis B activities [4,5].

Traditional cultivation of *Phyllanthus* species for medicinal purposes is carried out under field or greenhouse conditions, but is hindered by the low production of secondary metabolites of interest, such as lignans, as well as the natural dormancy of seeds, which limits its propagation aimed at large-scale cultivation [6].

Thus, plant tissue culture techniques could enable a more stable, homogeneous, and precise production system for control of plant development and the establishment of in vitro cultivation protocols that concentrate interesting secondary metabolites without the disadvantages of field cultivation [7].

Currently, in vitro culture of callus has been reported as an alternative method for the production of biomass aiming at the production of secondary metabolites, especially for the ease of induction and proliferation, being a simpler process than organogenic or embryogenic responses, which allows increasing the in vitro plant biomass, one of the main economic limitations in the production of secondary metabolites in vitro [8]. In addition, the induction and proliferation of calli in vitro allow good repeatability and standardization of responses [9].

In the process of induction and proliferation of calli, plant development regulators (PDRs) are used, where auxins and cytokinins are responsible for cell division and dedifferentiation [10] in tissues grown in vitro. Auxin 2,4-dichlorophenoxyacetic acid (2,4-D) is one of the most used for in vitro callus production, being highly efficient for the induction and proliferation of these calli, while its action can be intensified by the combination with 6-benzyladenine (BA) [11].

In vitro calli used for cell mass proliferation can be combined with chemical elicitors that stimulate specific secondary metabolite pathways, which can result in a high-performance technique in the study for production, with an increase in the yield of secondary metabolites under in vitro conditions [12]. Different studies on medicinal plants have shown good results using salicylic acid [13] and chitosan [14] as elicitors, activating different metabolic pathways and resulting in the increased concentration of some phenolic compounds in plants.

In addition, calli could be a possible source of new medicinal compounds scarcely reported in commonly differentiated tissues, such as leaves, roots, and stems of medicinal plants cultivated under field conditions. Thus, callus cell proliferation can be an efficient bioreactor for the biosynthesis of specific plant-derived medicinal compounds from relevant medicinal plants, such as *Phyllanthus* species. In this context, the main objective of this study was to develop a protocol for in vitro callus induction and proliferation in *Phyllanthus amarus*, aiming at the production of plant biomass and the identification of main secondary metabolites in the calli of this species, in comparison to those conventionally produced by micropropagated plants grown in greenhouses.

## 2. Results

The asepsis method used was successful in obtaining aseptic seedlings under in vitro conditions. Approximately 30% seeds germinated, resulting in seedlings after a period of 40–50 days of in vitro cultivation.

The transfer of nodal segments of these seedlings to Murashige and Skoog (MS) [15] culture medium containing 0.5 mg/L BA also successfully allowed the multiplication of healthy plants, with slight symptoms of vitrification in the leaves, but able to provide nodal segments for callus induction experiments and in vitro seedling production.

### 2.1. Experiments for In Vitro Callogenesis in Phyllantus amarus

Callus formation was observed firstly at the base of nodal segments (wound region) from the 14th day of cultivation, and the treatments with 2,4-D at 0.25 and 0.50 mg/L and without addition of BA presented 100% nodal segments containing calli. At the end of 60 days of cultivation, all nodal segments containing calli were also obtained in treatments with 0.25 mg/L and 0.50 mg/L 2,4-D, combined with 1.0 mg/L BA (Table 1; Figure 1).

In the treatments without the addition of phytoregulators, and in the treatments containing only 1 or 2 mg/L BA, without the addition of 2,4-D, there was no callus formation in darkness conditions (Table 1). In contrast, the addition of BA, even in dark conditions, resulted in the development of multiple shoots (Figure 1—T2 and T3).

In callus induction from nodal segments, only 2,4-D had an effect on the callus fresh weight, with no gains observed in the 2,4-D + BA combination. The treatment with 0.25 mg/L 2,4-D had the best result for fresh weight of calli (2.0 g/callus; >1.800% fresh weight gain), when compared with the initial explant fresh weight (Table 1).

For callus diameter, there were effects of 2,4-D, BA, and the interaction between these phytoregulators. The largest callus diameter (2.58 cm) was obtained in the treatment containing 2,4-D at 0.25 mg/L, without the addition of BA (Table 1; Figure 1—T4). The increase in the concentrations of 2,4-D (above 0.25 mg·L^−1^) or the addition of BA to 2,4-D treatments did not increase callus diameter (Table 1).

The results obtained in the callus proliferation phase showed that the concentrations of 0.50 and 0.75 mg/L 2,4-D, in the absence of BA, resulted in the largest callus diameters at 60 days of cultivation (2.04–2.39 cm). In the culture medium containing BA, 0.25 mg/L 2,4-D combined with BA (1 or 2 mg/L) produced the best results (2.03–2.17 cm), with no statistical differences between BA concentrations (Table 2). The results for fresh weight were very similar to those for diameter, with the largest increase in fresh weight of calli (2.30 g/callus) observed in the treatment with 0.75 mg/L 2,4-D (Table 2). The highest speeds in callus proliferation were also found in these treatments, mainly in treatments with 0.50 or 0.75 mg/L 2,4-D and 0.25 mg/L 2,4-D + 2.0 mg/L BA (data not shown).

Furthermore, calli of *P. amarus* successfully proliferated in vitro under light conditions using only BA as a phytoregulator (culture medium without 2,4-D). In these treatments, chlorophyll synthesis was observed in calli after the 15th day of cultivation, according to the acquisition of the green color and rigid consistency (Figure 2A). The best treatment for these conditions aiming at callus proliferation, determined by the fresh weight, was observed in the concentration of 2 mg/L BA (1.92 g/callus) (Table 3).

### 2.2. Chemical Elicitation of Calli Obtained in Medium with 2,4-D in the Dark and with BA in the Light

Treatments with salicylic acid (SA) and chitosan reduced the callus fresh weight only in dark conditions with 0.5 mg/L 2,4-D compared with the control. Under light conditions in culture medium containing BA, the fresh weight was not influenced by the elicitors (Table 4). In treatments with chitosan and SA in medium with 2,4-D under dark conditions, the calli obtained turned brown, unlike those obtained in the previous experiments and in the control (Figure 2D). Under light conditions and with addition of BA, there was an initial period of darkening of callus tissues with both elicitors, followed by the development of green and rigid calli, similar to those observed in the control.

Plants micropropagated (Figure 2B), acclimatized, and grown in pots in a greenhouse developed vigorously (Figure 2C), and the leaves were harvested for analysis at 60 days of cultivation. In these aerial samples from plants grown in a greenhouse, it was possible to observe eight different biosynthesized lignans, including 5-demethoxy-niranthin (402 g/mol), phyllanthin (418 g/mol), 5-demethoxy-nirtetralin (400 g/mol), nirtetralin (430 g/mol), niranthin (432 g/mol), and phyltetralin (416 g/mol). However, none of these lignans in plants were observed in the calli obtained in vitro, regardless of the culture conditions, light with BA or dark with 2,4-D, and the addition of chemical elicitors (Table 5).

Interestingly, calli grown in vitro produced other molecules not observed in plants grown in a greenhouse. In dark conditions, in a medium containing 2,4-D combined with the treatment with the salicylic acid (SA) elicitor, the GC-MS database was able to find 10 molecules, with the identification of squalene (410 g/mol), stigmasterol (412 g/mol), and γ-sitosterol (414 g/mol) (Table 5).

However, in calli obtained in BA treatment and cultivated under light conditions, the most representative molecule, observed independently of elicitors, was identified as betulin-aldehyde or betulone (440 g/mol), with relative percentage of 62.5% with control, 61% with SA, and 76% with chitosan. Stigmasterol and γ-sitosterol were also concentrated in all treatments. Two molecules, α-muurolene (204 g/mol) and β-amyrin (426 g/mol), were biosynthesized only in the control and with SA as an elicitor, but not with chitosan.

## 3. Discussion

The production of secondary metabolites by plants is the most economically viable way of producing these substances, mainly due to the complexity of these molecules, which makes their chemical synthesis very difficult [8,16].

A review published by Mao et al. [4] identified 514 molecules biosynthesized by different species of *Phyllanthus*, 37 of them in *P. amarus*, most of which were reported as having antitumor, anti-inflammatory, antiviral, antidiabetic, and hepatoprotective effects belonging to the group of lignans.

Under in vitro conditions, aiming at the production of secondary metabolites with greater control of the culture environment, a phase of tissue proliferation is established, which exponentially increases the number of cells to be used as bioreactors in the production of secondary metabolites, followed by an elicitation phase, in which some physical or chemical factor causing stress is applied directly to the cultivated tissues, aiming to concentrate these metabolites in the plants [8,17].

One of the biggest challenges in the in vitro production of secondary metabolites has been the efficiency of their production compared with the culture systems under field conditions. In this sense, studies are required to increase the concentration of these compounds in in vitro conditions [18], associated with a reduction in costs of micropropagation systems in order to make their production economically feasible under these conditions [8].

### 3.1. 2,4-D Was the Major Elicitor Responsible for Induction and Proliferation of Calli in Dark Conditions

Calli obtained from nodal segments with the addition of 2,4-D (without BA) in the culture medium had a beige color and a friable appearance, regardless of the concentration used in the culture medium (Figure 1). However, the largest amount of fresh weight and diameter of the calli were obtained at a concentration of 0.25 mg/L, and the addition of BA, in combination with 2,4-D, did not result in benefits in the induction phase. In the callus proliferation phase, the addition of BA (1 or 2 mg/L) was beneficial only at the concentration of 0.25 mg/L 2,4-D (Table 3 and Table 4).

Unander [19] also obtained friable calli induced from stem tissues of *Phyllanthus amarus*. These authors observed that the largest fresh weight of calli was obtained using the MS medium with 1.0 mg/L 2,4-D + 1.0 mg/L BA (0.68 ± 0.14 g/callus). In our study, the largest fresh weight (2.58 g/callus) of friable calli was also obtained with the ½ MS medium plus 2,4-D, but at a concentration of 0.25 mg/L, this being 3.8× higher than that obtained by Unander [19]. The increase in 2,4-D concentration or the addition of BA at 1.0 or 2.0 mg/L resulted in a gradual reduction in the callus fresh weight in the induction phase. Another difference found between that study and the present one was the cultivation conditions, whereby the current study was carried out in dark conditions, while that conducted by Unander [19] occurred under continuous lighting conditions.

As demonstrated in the present experiment, the addition of BA to 2,4-D treatments did not prove beneficial for the fresh weight and diameter of calli in the induction phase, compared with 2,4-D treatments. In another study, with *Phyllanthus amarus* and *P. urinaria*, there was successful induced callus production in nodal segments using MS medium, along with 7 µM 2,4-D + 1 µM BA. In addition, it was observed that the addition of cinnamic acid (30 µM), phenylalanine (1 µM), and naphthaleneacetic acid (1 µM) to this medium resulted in increments in the fresh weight of calli (1.5 g/callus, 1.3 g/callus, and 0.8 g/callus, respectively) for *P. amarus* [16].

In the proliferation phase of *P. amarus* calli, the largest fresh weights were obtained in concentrations of 0.25 mg/L + BA (1 or 2 mg/L) and 0.50 or 0.75 mg/L of 2.4-D without the addition of BA, with an interaction between these two phytoregulators (Table 4). The largest fresh weight of calli (2.30 g/callus) in the present study was obtained with 0.75 mg/L 2,4-D. Muthusamy et al. [16] using 7 µM 2,4-D + 1 µM BA reported fresh weight of 0.15–0.30 g/callus, much lower than values observed in the present study with the same species. Values close to 1.5 g/callus were reported by these authors only with the addition of 30 µM cinnamic acid.

### 3.2. The Use of Cytokinin BA, in the Presence of Light, Changed the Pattern of Callogenesis in P. amarus

An experiment was also conducted for proliferation of calli in the presence of light and subjected to different concentrations of BA added alone to the MS culture medium.

Although BA did not induce calli in nodal segments of *P. amarus* under dark conditions, its use alone induced high proliferation of calli under light conditions. Not only was there an increase in the fresh weight and diameter of calli, but there was also a change in the pattern of development of these calli, which changed from a beige color and a friable appearance (with 2,4-D + dark) to a green color and a hard consistency (BA + light), demonstrating that the action of this cytokinin is enhanced in the presence of light, especially since light has effects on the regulation of cytokinin synthesis, action, transport, and oxidation (Roman et al., 2016), generating changes in growth (cell divisions), as well as in the differentiation of callus cells, resulting in a green color (chlorophyll biosynthesis) and rigid consistency (biosynthesis of cell wall) (Figure 2A).

Despite these observed changes, callus proliferation remained high in these conditions (BA + light), with the largest gain in callus fresh weight at 2.0 mg/L (1.92 g/callus) (Table 3). Compared with previous studies on the proliferation of calli under similar conditions, the fresh weight obtained in the present experiment was 2.8× and 8.5× the fresh weight of calli obtained under similar conditions by Unander [19] and Muthusamy et al. [16], respectively, with *P. amarus.*

This demonstrates that the isolated use of 2,4-D in concentrations of 0.50–0.75 mg/L, with calli grown in dark conditions, or BA applied at 2.0 mg/L in the presence of light may result in an expressive gain in fresh weight or productivity of calli obtained in in vitro conditions in *P. amarus.* The proliferation rates in darkness with 0.75 mg/L and in light with 2.0 mg/L of BA were 22.1× and 12.0×, respectively.

The fresh weight gains reported herein are essential to make the technique of in vitro production of secondary metabolites viable from an economic point of view, with the advantages of in vitro cultivation and production of secondary metabolites from calli, such as the greater control of the production process, carrying out periodic harvests in shorter cycles independent of the time of the year in order to extract the plant-derived medicinal compounds (PDMC), the uniformity of response, and the absence of microbiological and chemical contaminations that affect the quality of PDMC produced and extracted under these conditions, when compared with the production of plants in agricultural systems [8,17].

### 3.3. The Use of Calli In Vitro Did Not Result in the Production of Phyllanthin, Even When Combined with Elicitors

The addition of the elicitors chitosan and salicylic acid reduced the fresh weight of calli in treatments with 2,4-D grown in the dark, but had no effect on the fresh weight of calli produced under light conditions in medium containing cytokinin BA.

In samples of plants grown in a greenhouse, it was possible to identify 5-demethoxy-niranthin lignans (Rt: 16.403 min, M: 402 g/mol), phyllanthin (Rt: 16.908 min, M: 418 g/mol), 5-demethoxy-nirtetralin (Rt: 17.336 min, M: 400 g/mol), nirtetralin (Rt: 18.202, M: 430 g/mol), niranthin (Rt: 19.854 min, M: 432 g/mol), and filtetralin (Rt: 16,91, M: 416 g/mol), as well as other unidentified molecules. Niranthin and nirtetralin, similar to phyllanthin and hypophyllanthin, also have an antiviral effect [3,19,20].

The use of gas chromatography was efficient and is, therefore, an effective technique for identifying lignans in plants of *P. amarus* (Table 5) [21] and cell suspension cultures of *P. niruri* [22]. Khan et al. [23] also showed that highest contents of phyllanthin in this species were found in leaves, followed by fruits and stems.

Despite the detection of lignans in aerial samples of plants grown in a greenhouse and from micropropagation, the calli obtained and proliferated in vitro of this same genotype did not concentrate any of the biosynthesized lignans observed in the plants, regardless of whether or not elicitors were added or the type of callus obtained with 2,4-D in the dark or with BA in the presence of light.

This response may be the result of the origin of the biosynthesis of these compounds. Lignans phyllanthin and hypophyllanthin are derived from lignins, polymers with phenylpropanoid units, which give rigidity to the cell wall, occurring mainly in the vascular tissues of plants, being formed only in differentiated cells with a well-developed secondary cell wall [24].

Considering that calli are almost entirely made up of cells in constant cell division and, therefore, undifferentiated, the non-detection of these lignans in calli in vitro may be a result of the stage of development of these cells. The friability of calli is an example of the presence of young cells. Friability is the capacity of plant cells to detach from each other; cells have meristematic characteristics, and friability is generally obtained in media with high concentrations of auxins [25], as demonstrated in the calli obtained in vitro for *P. amarus* in treatments with 0.25, 0.50, and 0.75 mg/L 2,4-D.

Even so, the results obtained here are opposite to those reported by Muthusamy et al. [16] with *P. amarus*, who observed phyllanthin lignan at very low concentrations (0.02–0.25 µg/g callus). These authors observed an increase in the concentration of phyllanthin in the calli in vitro with the addition of 30 µM of the amino acid phenylalanine. Elfahmi et al. [22] observed the production of the lignans urinatetralin and cubebin dimethyl ether in cell suspension cultures of *P. niruri*, not conventionally observed in plants, and the addition of 0.5 mM of ferulic acid or caffeic acid as precursors was beneficial to concentrate these compounds in the cells.

However, similar to the current study with *P. amarus*, phyllanthin, hypophyllanthin, niranthin, and nirtetralin lignans observed in the aerial samples of *P. niruri* plants were not identified in in vitro calli of the same species, where only 3,4-methylenedioxybenzyl-3′,4′-dimethoxybenzylbutyrolactone was observed [22].

Although Unander [19] did not quantify and characterize the secondary metabolites produced in vitro for different species of *Phyllanthus*, including *P. amarus*, this author found that the activity of the calli produced in vitro on viral enzymes was significantly lower than those obtained from extracts of whole plants grown in field conditions. Since lignans are largely responsible for the antiviral effect of *Phyllanthus* species extract, including *P. amarus* [26,27], it is understandable that the lower antiviral activity is associated with the low production capacity of these lignans in calli grown in vitro, when compared with extracts from plants grown in field conditions [19] or in a greenhouse (Table 5), as demonstrated in the present study, comparing the production of lignans in plants ex vitro vs. calli in vitro.

### 3.4. In Vitro Calli Are a Source of Other Metabolites of Great Medicinal Importance

Although the lignans produced in the aerial samples of plants grown in the greenhouse were not identified and observed in the induced and proliferated calli in vitro (Table 5), a greater diversity of different compounds was observed in the in vitro callus samples that were not observed in greenhouse plants, including compounds that were identified and others not available in the GC-MS database.

Quantitative and qualitative differences in metabolites produced in different parts of plants, calli, and cell suspension cultures were also observed in *P. niruri*, and in vitro cultures resulted in the identification of molecules not present in extracts collected directly from plants [22]. Anuar et al. [28] also detected the molecules quercetin and (+)-catechin in calli obtained in vitro from *P. niruri*, and the concentration of these molecules in the calli varied according to treatments with the phytoregulators Kinetin, 2,4-D, and 1-Naphthaleneacetic acid added to the culture medium.

The in vitro cultivation factors tested for calli, with 2,4-D in the dark or BA in the light, as well as the elicitors chitosan and salicylic acid, had effects on both the types of compounds and their abundance in the treatments.

As an example, for the sample of the treatment with the salicylic acid elicitor applied to the calli obtained with 2,4-D in dark conditions, the GC-MS database was able to find some compounds such as α-muurolene (Rt: 10.285, M: 204 g/mol), squalene (Rt: 25.039, M: 410 g/mol), stigmasterol (Rt: 29.112, M: 412 g/mol), and γ-sitosterol (Rt: 29.821, M: 414 g/mol), as highlighted in Table 5.

The compounds stigmasterol, γ-sitosterol, and a not identified (ni) molecule (422 g/mol) were found again in the green callus sample of the treatment with 2 mg/L BA with cultivation in the light with (salicylic acid or chitosan) or without (control) elicitation, similar to the calli obtained in the dark with 2.4-D (Table 5).

Moreover, there were identified molecules of differential occurrence between the types of callus observed in *P. amarus*. The presence of the ni molecules at 298, 314, 316, 392, and 394 g/mol and squalene (410 g/mL) was also verified in calli obtained in the dark + 2,4-D, while α-muurulen, β-amyrin, and other ni molecules were only observed in green calli, but not observed in calli obtained in the dark. The highest abundance of the molecules with molecular weights of 298, 300, and 422 g/mol was observed in calli in dark conditions with 2,4-D, while betulin-aldehyde or betulone (440 g/mol) showed the highest abundance presented in green calli derived from the BA treatment in light conditions.

From the molecules identified by GC–MS, some of these observed in the calli were not identified (ni) by GC-MS and (National Institute of Standards and Technology (NIST) databases (Table 5). An exhaustive and broad comparative search between these molecules of known molecular masses obtained from the calli was compared with those observed by different authors in other studies with *Phyllanthus* [4,20]. Molecules with the same molecular mass in these libraries were compared with the same molar weight of the metabolites observed in calli of *P. amarus*, and only the molecules betulin-aldehyde or betulone and β-amiiryn (Table 5) were identified via this method. This demonstrates that the calli are capable of synthesizing molecules very different from those already reported in the differentiated tissues of this species, as reported for the same genotype of *P. amarus* in this paper, some of which were reported for the first time in *P. amarus* and in the genus *Phyllanthus*.

From these comparisons, betulin or betulone and tetracyclic triterpenes were identified in green calli and were reported as the main metabolites with relative amounts ranging from 61–76% in green calli obtained with BA + light, from *P. amarus* (Table 5). These substances and some of their derivatives proved to be potent candidates with anticancer activity, showing cytotoxic activity against different human cell lines of carcinoma and in vitro stomach (MGC-803), breast (Bcap-37, MCF-7), prostate (PC3), melanoma (SK-MEL-2, A-375), medulloblastoma (Dayo), glioblastoma (LN-229), ovarian carcinoma (OVCAR-3) colon carcinoma (HT-29), and promyelocytic leukemia cancer cell lines (HL-60) [29,30]. Betulin and betulone could also be biotransformed [31] or synthetically modified to obtain new and more effective molecules against cancer cell lines [32,33].

## 4. Material and Methods

### 4.1. Phyllantus amarus Genetic Material and In Vitro Seeding

*Phyllanthus amarus* seeds from the germplasm collection of the Centro de Pluridisciplinar de Pesquisas Químicas, Biológicas e Agrícolas (CPQBA-Universidade Estadual de Campinas, Campinas, Brazil), known as CPQBA-14, were used due to their high concentrations of phyllanthin [6].

Seeds used as initial explants of *Phyllantus amarus* were subjected to asepsis with 70% alcohol for 30 s, followed by sodium hypochlorite solution (2.5% active chlorine) with two to three drops of Tween-20 for 20 min. Seeds were then immersed in sterile deionized water three consecutive times. After asepsis, they were germinated in MS medium [15] containing 1.5% sucrose and 0.8% agar, with pH adjusted to 5.8. Seeds were kept in a growth room with a 16 h photoperiod and approximately 20–25 μmol·m^−2^·s^−1^ photosynthetically active radiation provided by cold white fluorescent light, at 26 ± 1 °C, for a period of 10 weeks.

In vitro germinated seedlings were seeded on MS medium [15], containing half the concentration of macronutrients (MS ½), plus 2% sucrose, 0.1 g/L *myo*-inositol, 0.5 mg/L benzyladenine (BA), and 6.4 g/L agar until sufficient plants were obtained to carry out the experiments for callus induction.

### 4.2. Experiments for Induction and Proliferation of Calli in Nodal Segments of Phyllantus amarus

These experiments aimed to test the influence of 6-benzyladenine (BA) and 2,4-dichlorophenoxyacetic acid (2,4-D) on the induction of calli from stem nodal segments with approximately 1.0 ± 0.2 cm length and calli proliferation. For this, two experiments were carried out in the dark conditions, one for induction using the nodal segments of seedlings germinated in vitro and the second using the calli induced from the previous phase, with approximately 0.5 ± 0.2 cm diameter and 0.104 g/calli as explants. An experiment was also carried out with different concentrations of BA and callus cultivation under light conditions.

The culture medium used for the experiments was MS containing half the concentration of macronutrients (MS½), plus 3% sucrose and 0.1 g/L *myo*-inositol. The pH was adjusted to 5.8 and the medium was solidified with 6.4 g/L agar.

For the installation of the experiment, 12 treatments were used, in a factorial of 3 (BA) × 4 (2,4-D) with a completely randomized design, with 10 repetitions (test tubes) containing a nodal segment each. The treatments consisted of different combinations of BA (0, 1.0, and 2.0 mg/L) and 2,4-D (0, 0.25, 0.50, and 0.75 mg/L).

The culture media were poured into test tubes containing 12 mL medium each and autoclaved at 120 °C and 1 kgf/cm^2^, for 20 min.

The nodal segments were transferred to the growth room at 26 ± 1 °C in dark conditions for the entire experimental period.

For induction, the following were evaluated: the percentage of nodal segments induced to callus formation; weekly assessment of callus diameter in both the induction and the proliferation phase; the fresh biomass of calli, carried out at the end of 60 days of cultivation, in both phases of cultivation, with fresh weight measured on a precision analytical scale Mettler ML201 (Mettler-Toledo AG, Greifensee, Switzerland).

A third experiment was carried out with a part of the calli resulting from the previous experiment (2,4-D at 0.5 mg/L) transplanted to the MS ½ medium plus 3% sucrose and 0.1 g/L inositol. The pH was adjusted to 5.8 and the medium was solidified with 6.4 g/L agar, with the treatments given by different concentrations of BA at 0, 1.0, 2.0, 3.0, and 4.0 mg/L. The media were poured into 265 mL glass vials containing 35 mL of medium inoculated with five clusters of calli of 0.5 ± 0.2 cm each (repletion) and 0.16 g/callus. Calli were maintained in a 16 h photoperiod (25–30 μmol·m^−2^·s^−1^ cold fluorescent light) at 26 ± 1 °C, for 30 days.

### 4.3. Chemical Elicitation of Calli Obtained in Medium with 2,4-D in the Dark and with BA in the Light

For this experiment, calli obtained in dark (2,4-D) and light (BA) conditions were transplanted to the control medium, without elicitors, and to two treatments containing the elicitors salicylic acid [13] and chitosan [14], all with 20 test tubes containing one callus (repetition) each; the medium used was the MS ½ medium, plus 3% sucrose and 0.1 g/L inositol, supplemented with 2.0 mg/L BA. The pH was adjusted to 5.8 and the medium was solidified with 6.4 g/L agar.

In addition to the control, without the addition of elicitors, the effect of elicitors salicylic acid at 28 mg/L, added to the medium and cold sterilized with the aid of a 22 µm Millipore filter [13], and chitosan at 50 mg/L, added to the culture medium after autoclaving was verified. The chitosan solution was prepared by dissolution in 1 mL of glacial acetic acid at 55–60 °C for 15 min, and then the final volume was completed to 10 mL with distilled water and the pH adjusted to 5.8 with NaOH before its addition to the culture medium [14].

The experiment with the elicitors was maintained in a 16 h photoperiod (25–30 μmol·m^−2^·s^−1^ cold fluorescent light) at 25 ± 1 °C, for 45 days.

Evaluations of callus diameter were made weekly, and the fresh weight of calli was measured at the end of 35 days of cultivation. Phytochemical substances found in calli of *Phyllanthus amarus* were evaluated by GC-MS analysis, according to Section 4.5.

All experiments from Section 4.2 and Section 4.3 were tested by analysis of variance (ANOVA) and the Shapiro-Wilk normality test, followed by comparisons of means using the Scott-Knott test. The software used for all analysis was Agroestat (Barbosa and Maldonado, 2020, Jaboticabal, Brazil).

### 4.4. Cultivation of Micropropagated Plants in a Greenhouse

In order to compare the phytochemical profile of calli with that of plants grown in a greenhouse, GC-MS analyses were also performed on approximately 50 plants obtained from in vitro cultivation, on MS ½ medium without the addition of BA, with the objective of rooting and elongating in vitro, followed by acclimatization to substrate (peat + vermiculite + carbonized rice husk) in polypropylene trays. Plants were kept for 30 days in these conditions, planted in a plastic pot with 10 cm diameter, and fertigated weekly for the production of shoot biomass, aiming at the identification of compounds present in the aerial part of the plant.

### 4.5. GC-MS Analysis

Analyses were performed on a gas chromatograph (GC) Agilent HP-6890 with an HP-5MS capillary column (30 m × 0.25 mm × 0.25 μm), with the following working temperatures: injector 250°C, column 180 °C (5 min), 15 °C/min, 270 °C (15 min), 5 °C/min, 300 °C (10 min) and detector 300 °C. The chromatograph was coupled to an Agilent mass detector, HP-5975, operating at 70 eV, *m*/*z* = 30 to 500 u.m.a.

Two types of *P. amarus* extracts were prepared, one with callus samples, i.e., the four callus samples with three types of treatments for elicitation (control, salicylic acid, and chitosan), with BA at 2.0 mg/L, and another sample of dried plants from the aerial parts of plants grown in a greenhouse. All samples were dried for 24 h at 40 °C and stored at 8 °C maximum for 4 days, until the GC-MS analysis.

For the tests, 2 g (dry weight) of sample was weighed and transferred to a centrifuge tube with 10 mL of methanol. The sample was then crushed in a Polytron-Ultra Turrax 175 dispenser, model T-18 (Ika, Staufen, Germany) with a rotation of 3 = 16,000 rpm, for 1 min.

The extract was centrifuged for 6 min in a microprocessor centrifuge model 5500 D (Cientec, Piracicaba, Brasil) with a rotation of 300 rpm; the supernatant was transferred to a 25 mL volumetric flask, where 5 mL of methanol was added for the re-extraction of the compound, which was again centrifuged.

The supernatant was transferred to the 25 mL flask, the volume was completed with methanol, and a 6 mL aliquot of the sample was filtered through Millex and reserved for injection into the equipment. The sample was transferred to the injection vial and inserted into the equipment, which automatically injects 1.0 mL of sample into the detection column in the GC. The identification of molecules was performed using the NIST-11 library from the gas chromatograph Agilent^®^ (Agilent Technologies, Palo Alto, CA, USA), and from a specific library of lignans elaborated by CPQBA (Unicamp, Campinas, Brazil).

## 5. Conclusions

Our study provided the establishment of an efficient protocol for the proliferation of calli of *P. amarus* using 2,4-D in the dark or BA in the presence of light. Calli cultured in vitro also provided the production of new molecules not observed in differentiated tissues of greenhouse-grown plants of *P. amarus*, and this technique can be used for the production of molecules of interest for application as medicines, such as betulone. Some molecules, such as squalene and betulone, were not previously identified in the genus, while some phytosterols were previously identified in other species of *Phyllanthus* [4], with this being the first report for *P. amarus*. One of the biggest challenges in the production of secondary metabolites in vitro has been the high costs of the technique, compared to the conventional field production system of plant-derived medicinal compounds (PDMCs). However, the present study demonstrated the in vitro callus technique for the production of completely different PDMCs in *P. amarus* not produced by plants or parts of plants grown in the field, being of great potential for the discovery of new molecules, as well as for better comprehension of secondary metabolism routes aimed at the application of in vitro techniques for production and prospection of molecules of interest for medicinal purposes.

## Figures and Tables

**Figure 1 molecules-25-05895-f001:**
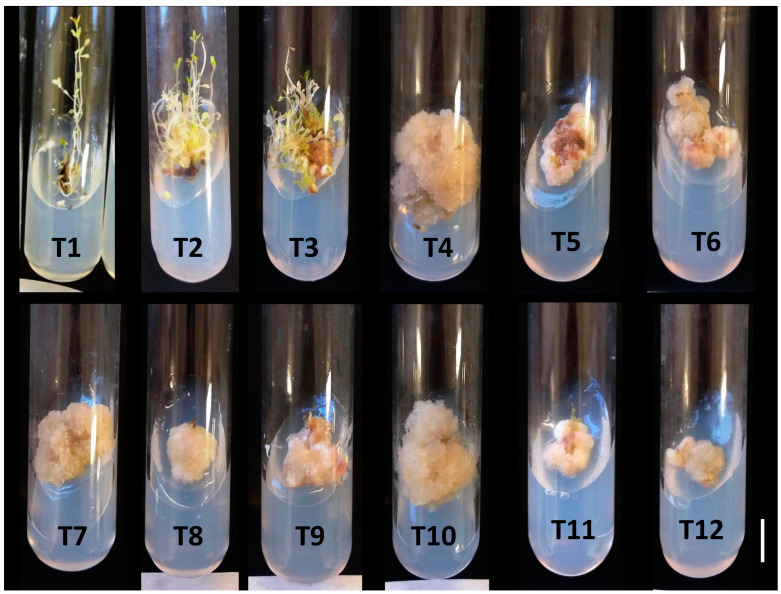
Differential responses of nodal segments of *Phyllanthus amarus* obtained with individual or mixed treatments with 2,4-dichlorophenoxyacetic acid (2,4-D) and 6-benzyladenine (BA) (mg/L): T1, free of phytoregulators (control); T2, BA 1.0; T3, BA 2.0; 0.25 mg/L 2,4-D (T4) + BA 1.0 (T5) or 2.0 (T6); 0.50 mg/L 2,4-D (T7) + BA 1.0 (T8) or 2.0 (T9); 0.75 mg/L 2,4-D (T10) + BA 1.0 (T11) or 2.0 (T12). White bar = 1.0 cm.

**Figure 2 molecules-25-05895-f002:**
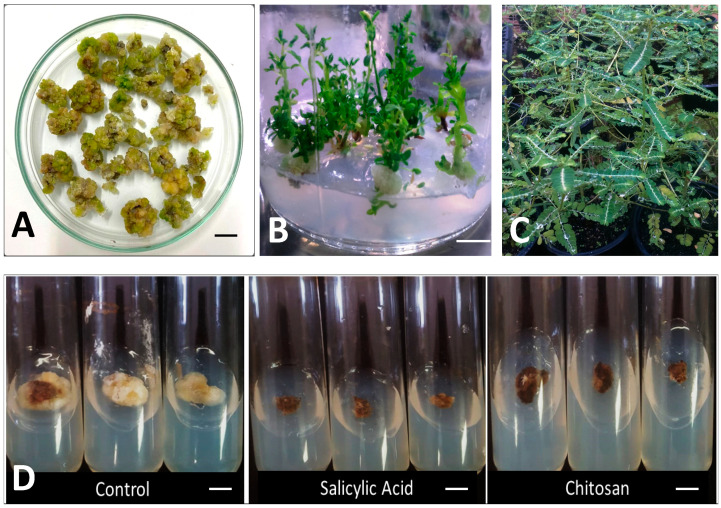
Green callus obtained with BA and lighting conditions (**A**); micropropagated plantlets under in vitro (**B**) and acclimatized under greenhouse conditions (**C**); effects of salicylic acid and chitosan on callus elicitation obtained with 2,4-D (0.50 mg/L) and darkness conditions (**D**) (bars = 1.0 cm).

**Table 1 molecules-25-05895-t001:** Effects of 2,4-dichlorophenoxyacetic acid (2,4-D) and 6-benziladenine (BA) on diameter (cm) and fresh mass weight (g/callus) of calli induced in nodal segments of *Phyllanthus amarus.*

2,4-D	Fresh Weight	BA/Callus Diameter (cm) Mean ± SD		
(mg/L)	(g/callus) Mean ± SD	0 mg/L	1 mg/L	2 mg/L
0	0.00 ± 0.00 ^c^	0.00 ± 0.00 ^cA^	0.00 ± 0.00 ^cA^	0.00 ± 0.00 ^bA^
0.25	2.00 ± 0.23 ^a^	2.58 ± 0.30 ^aA^	2.00 ± 0.30 ^aB^	1.76 ± 0.37 ^aB^
0.50	1.41 ± 0.30 ^b^	2.23 ± 0.37 ^bA^	1.59 ± 0.21 ^bB^	1.57 ± 0.04 ^aB^
0.75	1.24 ± 0.20 ^b^	1.99 ± 0.58 ^bA^	1.43 ± 0.20 ^bB^	1.56 ± 0.29 ^aB^
F (2,4-D) ^1^	103.07 **	294.05 **		
F (BA)	5.57 **	31.10 **		
F (2,4-D × BA)	0.94 ^ns^	4.39 **		
CV (%)	11.55	21.73		

^1^ Each mean represents 10 replicates (test tubes) with one nodal segment each. Means within a column (2,4-D effects; lowercase) or lines (BA effects; uppercase) followed by different letters are significantly different according to the Scott-Knot test at 1% (**) level of probability. All data followed a normal distribution according to the Shapiro-Wilk test. SD = standard deviation of means; F = distribution of variances according the F test; CV = coefficient of variation; ns = not significant.

**Table 2 molecules-25-05895-t002:** Effects of 2,4-dichlorophenoxyacetic acid (2,4-D) and 6-benziladenine (BA) on diameter (cm) and fresh mass weight (g/callus) of proliferated calli of *Phyllanthus amarus*.

2,4-D	BA			BA		
(mg/L)	0 mg/L	1 mg/L	2 mg/L	0 mg/L	1 mg/L	2 mg/L
	Callus Diameter (cm) Mean ± SD			Fresh Weight (g) Mean ± SD		
0.00	1.01 ± 0.23 ^cA^	1.11 ± 0.36 ^bA^	1.09 ± 0.41 ^dA^	0.17 ± 0.31 ^bA^	0.32 ± 0.24 ^bA^	0.30 ± 0.41 ^bA^
0.25	1.05 ± 0.26 ^cB^	2.03 ± 0.26 ^aA^	2.17 ± 0.56 ^aA^	0.16 ± 0.37 ^bB^	1.79 ± 0.34 ^aA^	2.01 ± 0.24 ^aA^
0.50	2.39 ± 0.28 ^aA^	1.85 ± 0.29 ^aB^	1.73 ± 0.31 ^bB^	1.89 ± 0.44 ^aA^	1.29 ± 0.28 ^aA^	1.37 ± 0.44 ^aA^
0.75	2.04 ± 0.41 ^bA^	1.66 ± 0.58 ^aB^	1.50 ± 0.17 ^cB^	2.30 ± 0.33 ^aA^	1.15 ± 0.38 ^aB^	0.86 ± 0.31 ^bB^
F (2,4-D) ^1^	35.30 **				31.97 **	
F (BA)	0.16 ^ns^				0.31 ^ns^	
F (2,4-D × BA)	14.35 **				14.34 **	
CV (%)	24.39				14.34	

^1^ Each mean represents 10 replicates (test tube) with one nodal segment each. Means within a column (2,4-D effects; lowercase) or lines (BA effects; uppercase) followed by different letters are significantly different according to the Scott-Knott test at the 1% (**) level of probability. All data followed a normal distribution according to the Shapiro-Wilk test. SD = standard deviation of means; F = distribution of variances according the F test; CV = coefficient of variation; ns = not significant.

**Table 3 molecules-25-05895-t003:** Fresh weight of calli of *Phyllanthus amarus* proliferated in vitro under concentrations of 6-benziladenine.

BA (mg/L) ^1^	Fresh Weight (g/callus) Mean ± SD
0.0	1.19 ± 0.44 ^d^
1.0	1.50 ± 0.31 ^c^
2.0	1.92 ± 0.46 ^a^
3.0	1.76 ± 0.38 ^b^
4.0	1.69 ± 0.33 ^b^
F	133.74 **
CV (%)	3.36

^1^ Each mean represents 10 replicates (test tube) with one nodal segment each. Means within a column followed by the different letters are significantly different according to the Scott-Knott test at the 1% (**) level of probability. All data followed a normal distribution according to the Shapiro-Wilk test. F = Distribution of variances according the F test; CV = coefficient of variation; SD = standard deviation of means.

**Table 4 molecules-25-05895-t004:** Effects of the elicitors chitosan and salicylic acid (SA) on in vitro callus growth under darkness + 2,4-D and light + 6-BA.

Elicitors ^1^	0.5 mg/L 2,4-D + Darkness		2.0 mg/L BA + Light
	Diameter (cm) Mean ± SD	Fresh Weight (g/callus) Mean ± SD	Fresh Weight (g/callus) Mean ± SD
Control	0.50 ± 0.24 ^a^	1.51 ± 0.45 ^a^	1.78 ± 0.41 ^a^
Salicylic acid	0.35 ± 0.38 ^a^	0.90 ± 0.38 ^c^	1.83 ± 0.33 ^a^
Chitosan	0.40 ± 0.44 ^a^	1.31 ± 0.36 ^b^	1.77 ± 0.48 ^a^
F ^1^	0.33 ^ns^	24.59 **	0.07 ^ns^
CV (%)	13.17	15.89	21.16

^1^ Each mean represents 10 replicates (test tube) with one nodal segment each. Means within a column followed by the different letters are significantly different according to the Scot-Knott test at the 1% (**) level of probability. All data followed a normal distribution according to the Shapiro–Wilk test. F = distribution of variances according the F test; CV = coefficient of variation; SD = standard deviation of means; ns = not significant.

**Table 5 molecules-25-05895-t005:** Biomolecules biosynthesized by greenhouse plants and in vitro types of calli of *Phyllanthus amarus* under different conditions and identified by GC-MS.

Rt (min)	Formula	Molar Weight	Biomolecule	Greenhouse Plants	Callus				References
					BA + Light (Relative Content %)			2,4-D + Dark	
		(g·mol^−1^)		(Leaves + Stem)	Control	Salicylic Acid	Chitosan	Salicylic Acid	
10.86	C_15_H_24_	204	alpha-muurolene		1.33	2.23			NIST-11
			ni				0.61		
		298	ni					13.90	
		300	ni		3.97	2.49		14.69	
		314	ni					0.81	
		316	ni					1.50	
		386	ni	3.15%					
		392	ni					0.50	
		394	ni					0.59	
16.40	C_23_H_30_O_6_	402	5-demethoxy-niranthin	23.34%					Library CPQBA
16.91	C_24_H_34_O_6_	418	Phyllanthin	30.03%					Library CPQBA
17.02	C_24_H_32_O_6_	416	Phyltetralin	6.19%					Library CPQBA
17.34	C_23_H_28_O_6_	400	5-demethoxy-nirtetralin	10.01%					Library CPQBA
18.26	C_24_H_30_O_7_	430	Nirtetralin	3.07%					Library CPQBA
19.85	C_24_H_32_O_7_	432	Niranthin	18.92%					Library CPQBA
24.96	C_29_H_48_O	412	Stigmasterol		5.22	8.14	5.45	8.02	NIST-11
25.04	C_30_H_50_	410	Squalene					0.73	NIST-11
26.86	C_29_H_50_O	414	γ-Sitosterol		2.56	3.35	5.14	6.72	NIST-11
28.42	C_30_H_50_O	426	β-Amyrin		2.58	2.57			NIST-11
		422	ni		9.93	8.95	7.90	29.10	
			ni						
			ni		4.74	4.57	2.60		
			ni			2.34			
			ni			6.2			
32.67	C_30_H_48_O_2_	440	Betulin aldehyde or betulone		62.5	60.92	75.98	4.69	NIST-11
			ni		4.29		2.32		
			ni		2.88				
		Total Molecules		7	10	10	7	11	

ni, not identified molecules presented in calli and detected by GC-MS; NIST-11, National Institute of Standards and Technology Library; CPQBA, Centro Pluridisciplinar de Pesquisas Químicas, Biológicas e Agrícolas.

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
