# Peer review of "In Vitro Calli Production Resulted in Different Profiles of Plant-Derived Medicinal Compounds in Phyllanthus amarus"

_molecules, 2020, doi:10.3390/molecules25245895_

Round 1

Reviewer 1 Report

This research answers the question of whether plant material can be produced "in vitro" in the species Phyllanthus amarus. This has been shown to be successful.
The authors justified this research by the fact that the necessary biomass from Phyllanthus amarus is not sufficient from field crops or spontaneous flora. In addition it is the problem of difficult seed germination.

The subject has a medium to good degree of originality. Compared to existing studies, the authors of this paper add new aspects regarding in vitro callogenesis at Phyllantus amarus and use of different substances for chemical elicitation of callus.

Yes, the work is accessible and relatively easy to read and understand.

The conclusions are in line with the evidence and arguments presented. Is presented an efficient protocol for the proliferation of calli of Phyllantus amarus usind 2,4-dichlorophenoxyacetic acid and 6-benzyladenine. Thus responding to the purpose of the research.

Minor typos:

row 208: ” phase of P. amaru scalli,”

348        (CBPQA-UniversidadeEstadual de Campinas)

350        amaruswere

420-422    30m x 0.25mm x 0.25µm;    5min        (space is left after numbers)

It would be useful to assess the cost difference between in vitro production compared to free field production.

Author Response

Dear Reviewer 1:

 We like to thanks your efforts and contributions to our manuscript. All suggestions and corrections appointed were realized in the manuscript. 

  About the production costs, we prefer do not add because in vitro callus don't produced the same biochemical profile of plants under greenhouse cultivation, difficulting these comparisons with precise information. Thus, in vitro callus produced molecules that do not could produced and identified in the plants, as well as, lignans was not produced under in vitro conditions. We try to find some paper about production costs of Phyllanthus under field conditions, but no reports were found. 

  AHowever, to provide better explanation about the main novelty of the paper, associated with costs, we improved the conclusion section emphasizing this question, showing the importance of in vitro systems to prospect and improve production of secondary metabolites in vitro.

Many thanks and best regards. Jean Cardoso

Reviewer 2 Report

In “In vitro calli production resulted in different profile of plant-derived medicinal compounds in Phyllantus amarus”, the authors used an in vitro technique aimed at producing plant fresh weight and plant-derived medicinal compounds (PDMC) through the induction and proliferation of calli from Phyllanthus amarus. Then the authors compared the contents of PDMC in calli with those present in in-vitro grown plants.

Table 1, 2, 3 and 4. The authors should insert the statistical differences (letters) as apex in all the tables.

Line 126. “chlorophyll synthesis was verified...”. Did the authors use a specific method to determine the content of chlorophyll in calli? If yes, please explain this in the materials and methods section.

Figure 1 and 2. May the authors add a scale bar in these Figures?

4.5 GC-MS analysis. Did the authors use standards to identify and measure the compounds of interest? If yes, details of the standards used should be included in this paragraph.

Line 429. Did the authors crush the samples under low temperature? Usually in this type of analysis low temperatures are necessary to avoid molecules’ degradation.

A much more detailed description of the statistical analyses used should be included. The authors should add a little paragraph gathering all the pieces of information relative to the assays/programs used to calculate the statistical differences among the samples. For example, did the authors check the homogeneity and normal distribution of the data? Did the authors apply ANOVA for each set of data?

Table 5 should be improved by proving more pieces of information about the compounds identified with GC-MS, namely the retention time, formula, theoretical mass, observed mass, error mass and references/standards used for the identification per each compound detected.

The conclusion section should be amplified emphasizing the aspects of novelty present in this study.

Please revise the English language, since some mistakes and typos are present in the main text.

The manuscript addresses an interesting topic; however, some technical aspects are lacking and others remain unclear, especially in the materials and methods section. Furthermore, in my opinion, the manuscript lacks novelty. The authors should highlight the aspects which make this study innovative, in particular in the discussion and conclusions sections.

Author Response

Dear Reviewer 2, Thank you very much for your detailed and careful revision. Main changes in the manuscript was described below and detailed in blue text in the manuscript. Jean Cardoso

In “In vitro calli production resulted in different profile of plant-derived medicinal compounds in Phyllantus amarus”, the authors used an in vitro technique aimed at producing plant fresh weight and plant-derived medicinal compounds (PDMC) through the induction and proliferation of calli from Phyllanthus amarus. Then the authors compared the contents of PDMC in calli with those present in in-vitro grown plants.

Table 1, 2, 3 and 4. The authors should insert the statistical differences (letters) as apex in all the tables.

Response: These changes were realized in all Tables containing statystical analysis. Also, we improved the footnote of tables for more detailed information of statystical analysis.

Line 126. “chlorophyll synthesis was verified...”. Did the authors use a specific method to determine the content of chlorophyll in calli? If yes, please explain this in the materials and methods section.

Response: Chlorophyll synthesis was not verified in calli, but only observed visually in the callus by the acquisition of green color. We changed this sentence for better comprehension, as follow:

...chlorophyll synthesis was observed in the callus after the 15th day of cultivation, by the acquisition of the green color and rigid consistency (Figure 2A).

Figure 1 and 2. May the authors add a scale bar in these Figures?

Response: Scale bars were added in the Figures.

4.5 GC-MS analysis. Did the authors use standards to identify and measure the compounds of interest? If yes, details of the standards used should be included in this paragraph.

Response: There no used analytical Standards. We improved this part in the material and methods section about how was identified the molecules, as follow:

The identification of molecules were realized using NIST-11 library from the gas chromatograph Agilent®, and from a specific library of lignans elaborated by CPQBA (Unicamp, Campinas, Brazil)

Line 429. Did the authors crush the samples under low temperature? Usually in this type of analysis low temperatures are necessary to avoid molecules’ degradation.

Response: There were used dry weight for the analysis, as recommended by previous references with this species. Due to the characteristic of the samples, low temperatures would not allow the elution of the analytes through the chromatographic column. No signs of degradation of the molecules were detected.

In addition, we added information about the temperature and time for drying samples and storage conditions before analysis in the material and methods. The methodology used was based on the follow reference (Vaz et al. 2006) and cited in the manuscript.

A much more detailed description of the statistical analyses used should be included. The authors should add a little paragraph gathering all the pieces of information relative to the assays/programs used to calculate the statistical differences among the samples. For example, did the authors check the homogeneity and normal distribution of the data? Did the authors apply ANOVA for each set of data?

Response: These information were added in lines 408-410 of the manuscript, as follow:

‘All experiments from itens 2.2 and 2.3 were submitted to Analysis of Variance (ANOVA) and Shapiro-Wilk normality test, followed by comparisons of means using Scott-Knott test. The software used for all analysis was Agroestat (Barbosa and Maldonado, 2020).’

For homogeneity we used Shapiro-Wilk test. Also, in each table we added informations about this test and the results of analyzed datas.

We was also provided more detailed information in table footnote about the analyzed data and type of analysis, as follow:

1Each mean represents ten replicates (test tubes) with one nodal segment each. Means within a column (2,4-D effects; lowercase) or lines (BA effects; uppercase) followed by different letters are significantly different based on Scott-Knot test at the 5% (*) or 1% (**) level of probability. All data followed the normal distribution according Shapiro-Wilk test. St. dev. = Standard deviation of means. F = Distribution of variances according the F test, CV = Coeficient of variation.

Table 5 should be improved by proving more pieces of information about the compounds identified with GC-MS, namely the retention time, formula, theoretical mass, observed mass, error mass and references/standards used for the identification per each compound detected.

Response: The table 5 was improved by addition of the informations requested.

The conclusion section should be amplified emphasizing the aspects of novelty present in this study.

Response: We amplified and emphasing the main novelty of the present study in the conclusion, as follow: One of the biggest challenges in the production of secondary metabolites in vitro has been the high costs of the technique, compared to the conventional field production system of plant-derived medicinal compounds (PDMCs). However, the current study demonstrated the great potential of the in vitro callus production technique aiming at the production of completely different PDMCs and not produced by plants or parts of plants grown in the field, being of great potential in the discovery of new molecules, production routes and applications aimed at application of in vitro techniques in the production and prospection of molecules of interest for medicinal purposes.

Please revise the English language, since some mistakes and typos are present in the main text.

Response: The english language was completely and carefully revised by authors and also by specialized editing service.

The manuscript addresses an interesting topic; however, some technical aspects are lacking and others remain unclear, especially in the materials and methods section. Furthermore, in my opinion, the manuscript lacks novelty. The authors should highlight the aspects which make this study innovative, in particular in the discussion and conclusions sections.

Reviewer 3 Report

The study presents a method optimized for production of high biomass of the medicinal plant Phyllanthus amarus in vitro. A subsequent elucidation of the tissue composition revealed absence of the lignans and, on the other hand, presence of other potentially interesting biomolecules. Generally, in my opinion, the study is interesting and well done. However, several aspects can be improved:

  1. I recommend a language check. There are several spelling mistakes (e.g., in the title, change Phyllantus to Phyllanthus) and also some sentences that are not completely clear (lines 42-44, 45-48, 62, 107-108, 125, 159, 171, 211, 217, 328, 332, 337). Further, please unify calluses/calli and the unit signs.
  2. In Tables 1-4, you state that means followed by the same letter are not significantly different. Are the means followed by different letters significantly different? Does the Scott-Knott test provide this kind of information? If not, I recommend another statistical test. Also, please explain the meanings of “F“ and “CV“ in the footers.
  3. In Table 1, the weight data correspond to which level of BA? In Table 4, why was the diameter not measured under light conditions?
  4. Line 107, do you really want to refer to Fig 2A? (The treatments do not match.)
  5. Why did you chose the used concentrations of elicitors? Did you try more different concentrations?
  6. Table 5, the analytes are quantified as relative content in %. Do you mean relative content in the extract? Would it be possible to express the contents for example in µg/g FW of the tissue? It would facilitate the comparison with other studies. Further, I did not find the “ha“ from the footer in the table.
  7. In the Discussion section, there are some references that have a different format and are not mentioned in the list of references (e.g., Teramoto et al., Khan et al., and others).
  8. L 202, not clear, which authors do you mean.
  9. L 357-359, I suppose there was agar in the medium?
  10. L 363, you state there were two experiments in dark conditions. L 371, the setup for the first experiment is provided (induction?)? Or for both experiments (induction and proliferation?)? Please clarify.
  11. L 423, “operating at 70 and V”? Please clarify.
  12. L 428, please add the conditions (temperature) of drying.

Author Response

Reviewer 3 -

Firstly we like to thanks your several and careful revision of our manuscript. We hope that we provide all changes you requested. Thanks and best regards. Jean Cardoso

Comments and Suggestions for Authors

The study presents a method optimized for production of high biomass of the medicinal plant Phyllanthus amarus in vitro. A subsequent elucidation of the tissue composition revealed absence of the lignans and, on the other hand, presence of other potentially interesting biomolecules. Generally, in my opinion, the study is interesting and well done. However, several aspects can be improved:

  1. I recommend a language check. There are several spelling mistakes (e.g., in the title, change Phyllantus to Phyllanthus) and also some sentences that are not completely clear (lines 42-44, 45-48, 62, 107-108, 125, 159, 171, 211, 217, 328, 332, 337). Further, please unify calluses/calli and the unit signs.

Response: All these sentences were revised and rewritten for better comprehension. The language was revised completelly by an editing service.

  1. In Tables 1-4, you state that means followed by the same letter are not significantly different. Are the means followed by different letters significantly different? Does the Scott-Knott test provide this kind of information? If not, I recommend another statistical test. Also, please explain the meanings of “F“ and “CV“ in the footers.

Response: These parts were rewritten for more precise and especific comprehension of the test. Meanings of F and CV was also added in the table footnote.

  1. In Table 1, the weight data correspond to which level of BA? In Table 4, why was the diameter not measured under light conditions?

Response: According the ANOVA test , there are no effects of interaction between BA x 2,4-D. Thus, there are presented data from the means of different levels of BA for each 2,4-D concentration. By our previous results, diameter and fresh weight are highly correlated. Thus, in the last experiment, there was measured only fresh weight of calli.

  1. Line 107, do you really want to refer to Fig 2A? (The treatments do not match.)

Response: This sentence, as well as, the reference to Fig 2a was deleted from the text.

  1. Why did you chose the used concentrations of elicitors? Did you try more different concentrations?

Response: Concentrations was based on different references previous published and used as elicitor for in vitro secondary metabolites production (13 and 14), as follow:

  1. Coste, A.; Vlase, L.; Halmagyi, A.; Deliu, C.; Coldea, G. Effects of plant growth regulators and elicitors on production of secondary metabolites in shoot cultures of Hypericum hirsutum and Hypericum maculatum. Plant Cell. Tissue Organ Cult.2011, 106, 279–288, doi:10.1007/s11240-011-9919-5.
  2. Zhao, J.L.; Zhou, L.G.; Wu, J.Y. Effects of biotic and abiotic elicitors on cell growth and tanshinone accumulation in Salvia miltiorrhiza cell cultures. Appl. Microbiol. Biotechnol.2010, 87, 137–144, doi:10.1007/s00253-010-2443-4.

  1. Table 5, the analytes are quantified as relative content in %. Do you mean relative content in the extract? Would it be possible to express the contents for example in µg/g FW of the tissue? It would facilitate the comparison with other studies. Further, I did not find the “ha“ from the footer in the table.

Response: Due to the high costs for quantification and high number of metabolites identified, we attempted our study on identification of differential production of metabolites obtained from differentiated tissues (greenhouse plants) x in vitro callus, the most novelty of this study. Thus, we not measured the total amount in µg/g. However, we added informations about the molecules obtained, improving the table 5. The term ‘ha’ was replaced for the numerical relative amount in the samples, for better quality of results. The term ‘ha’ was deleted from the footnote.

  1. In the Discussion section, there are some references that have a different format and are not mentioned in the list of references (e.g., Teramoto et al., Khan et al., and others).

Response: All references were revised and formatted according Molecules journal instructions

  1. L 202, not clear, which authors do you mean.

Response: This sentence was rewritten for better comprehension.

  1. L 357-359, I suppose there was agar in the medium?

Response: Yes, we completed these informations in the sentence.

  1. L 363, you state there were two experiments in dark conditions. L 371, the setup for the first experiment is provided (induction?)? Or for both experiments (induction and proliferation?)? Please clarify.

Response: We completed these information in this sentence, as follow: ‘For the installation of both experiments (induction and proliferation of calli), 12 treatments...’

  1. L 423, “operating at 70 and V”? Please clarify.

Response: These informations was corrected, as follow 70 eV, mz= 30 to 500 u.m.a.

  1. L 428, please add the conditions (temperature) of drying.

Response: Added as follow: All samples were dried for 24-h at 40°C and stored at 8°C maximum for 4-days, until the GC-MS analysis.

Round 2

Reviewer 2 Report

The authors address all the questions and comments I pointed out in the first round of revision using appropriate explanations. Furthermore, the quality of the manuscript has been substantially improved. I am therefore satisfied with the revisions.

Author Response

have already been revised
